# Orthostatic hypotension and neurocognitive disorders in older women: Results from the EPIDOS cohort study

Guillaume T. Duval[1,2]*, Anne-Marie Schott[3,4], Yves Rolland[5], Jennifer Gautier[1], Hubert Blain[6], Gustavo Duque[7,8], Cedric Annweiler[1,2,9]

1 Department of Geriatric Medicine, University Memory Center, Research Center on Autonomy and Longevity (CeRAL), Angers University Hospital, Angers, France, 2 School of Medicine and UPRES EA 4638, University of Angers, Angers, France, 3 Research on Healthcare Performance (RESHAPE), INSERM U1290, Université Claude Bernard Lyon 1, Lyon, France, 4 Hospices Civils de Lyon, Pôle de Santé Publique, Service de Recherche et D'épidémiologie Cliniques, Lyon, France, 5 Department of Geriatrics, Toulouse University Hospital, INSERM U1027, University of Toulouse III, Toulouse, France, 6 Department of Internal Medicine and Geriatrics, Montpellier University Hospital, University of Montpellier 1, Montpellier, France, 7 Australian Institute for Musculoskeletal Science (AIMSS), The University of Melbourne and Western Health, St. Albans, Victoria, Australia, 8 Department of Medicine, Melbourne Medical School–Western Precinct, The University of Melbourne, St. Albans, Victoria, Australia, 9 Robarts Research Institute, Schulich School of Medicine and Dentistry, The University of Western Ontario, London, Ontario, Canada

* guillaume.duval@chu-angers.fr

**Data Availability Statement:** All relevant data are within the manuscript and its Supporting Information files.

## Abstract

### Background

Although it is well-admitted that cardiovascular health affects cognition, the association between orthostatic hypotension (OH) and cognition remains unclear. The objectives of the present study were i) to determine among the EPIDOS cohort (EPIdémiologie de l'OStéoporose) whether OH was cross-sectionally associated with cognitive impairment at baseline, and ii) whether baseline OH could predict incident cognitive decline after 7 years of follow-up.

### Methods

Systolic and Diastolic Blood Pressure (SBP and DBP) changes while standing (ie, ΔSBP and ΔDBP, in %) were measured at baseline among 2,715 community-dwelling older women aged 75 years and older using no antihypertensive drugs from the French EPIDOS cohort. OH was defined as a decrease in SBP ≥20 mmHg and/or a decrease in DBP ≥10 mmHg within 3 min after standing. Cognitive impairment was defined as a Short Portable Mental Status Questionnaire (SPMSQ) score <8 (/10). Among those without cognitive impairment at baseline, a possible incident onset of cognitive decline was then sought after 7 years of follow-up among 257 participants.

### Results

Baseline ΔSBP was associated with baseline cognitive impairment (adjusted OR = 1.01, p = 0.047), but not with incident onset of cognitive decline after 7 years (adjusted OR = 0.98, p =

**Funding:** The authors received no specific funding for this work.

**Competing interests:** The authors have declared that no competing interests exist.

0.371). Neither baseline OH nor baseline ΔDBP were associated with cognitive impairment neither at baseline (p = 0.426 and p = 0.325 respectively) nor after 7 years (p = 0.180 and p = 0.345 respectively).

## Conclusions

SBP drop while standing, but neither OH *per se* nor DBP drop while standing, was associated with baseline cognitive impairment in older women. The relationship between OH and cognitive impairment appears more complex than previously expected.

## Introduction

Cognitive decline is a worldwide major health concern in the aging population because of its adverse consequences and its expanding prevalence and incidence [1]. In order to reduce the impact at an individual level and in terms of health and social costs, the development of efficient therapeutic strategies proves necessary. Regrettably, the development of all current specific medications is unsuccessful in clinical development phases [2, 3]. However, an interesting fact is that the prevalence of major neurocognitive disorders in the general population might still be subject to changes. Indeed, a reduction in the prevalence of dementia is reported among English community-dwellers over the past two decades [4]. So, even if some factors, such as the increased life expectancy beyond 80 years, are unavoidable and augment the global prevalence of neurocognitive disorders, others may instead reduce its prevalence. For instance, efficient primary prevention of cardio-vascular disease is suspected to explain the reduced prevalence of dementia in England [4]. For instance, both ischemic and hemorrhagic strokes are reported among the major causes of dementia. Similarly, blood pressure abnormalities have also been associated to cognitive decline and brain changes, including a greater burden of white matter hyperintensities and a higher degree of brain atrophy [5–7]. Hypertension may also result in greater risk of cognitive disorders due to atherosclerosis and subsequent cerebral hypoperfusion [5]. In contrast, the possible role of orthostatic hypotension (OH) has not yet been elucidated.

Orthostatic hypotension, which is defined as a drop of blood pressure within 3 minutes after the switch to orthostatism, affects about 5% of general population before 50 years old and its prevalence increases with advance in age, affecting up to 30% of people over 70 years old [8]. The clinical relevance is that OH leads to major blood pressure variability with low brain flow pressure as a consequence, and greater risks of syncope, falls and mortality [8]. Based on the adverse consequences of blood pressure abnormalities, and specifically of low brain flow pressure, on brain health and function [9], we hypothesized that OH may result in cognitive decline among older adults. The objectives of the present analysis were to determine among the EPIDOS (EPIdémiologie de l'OStéoporose) cohort study i) whether OH was cross-sectionally associated with cognitive impairment at baseline assessment, and ii) whether baseline OH could predict incident cognitive decline after 7 years of follow-up.

## Methods

### Participants

**EPIDOS study.** We studied community-dwelling older women from the EPIDOS study, a French observational prospective multicentric national cohort designed to evaluate the risk

factors for hip fracture among community-dwelling older women. Sampling and data collection procedures have been described in detail elsewhere [10]. In summary, from 1992 to 1994, 7,598 women aged 75 years and older were recruited from electoral lists in five French cities (Amiens, Lyon, Montpellier, Paris and Toulouse). All included study participants received at baseline a full medical examination by trained nurses in each local clinical center, which consisted of structured questionnaires and a clinical examination. For the present analysis, we used as exclusion criterion the use of antihypertensive medication at baseline, defined as diuretics, beta-blockers, calcium antagonists, angiotensin conversion enzyme inhibitors, agonists of angiotensin-II receptors or central antihypertensive agents (n = 4,662). We also excluded 60 women due to missing data regarding OH, and 161 women due to missing data regarding the other selected variables. Finally, 2,715 women met the selection criteria and had all available data at baseline. Fig 1 shows the flowchart of the study.

**Toulouse cohort study.** At the end of the EPIDOS 4-year study, all participants included in the centre of Toulouse were invited to take part in an additional 3-year follow-up study. The data collection procedures and flow diagram have been described elsewhere in detail [11]. In summary, all women who had given informed consent were offered a consultation exactly seven years after their inclusion. They were assessed either at home or at the Department of Internal Medicine and Clinical Geriatrics of Toulouse University Hospital. Initially, 1,462 women were included in the EPIDOS study in Toulouse. At the end of the 7-year follow-up, data on cognitive status (i.e., no dementia, Alzheimer disease, or other types of dementia) were available for 714 women (48.8% of the initial cohort). Of the other 748 women whose cognitive status remained undetermined, 193 (25.8%) died during follow-up, 414 (54.7%) were lost to follow-up, and 141 (18.9%) withdrew from the follow-up study [11]. To focus solely on the association of OH with cognitive function, women were excluded from the longitudinal

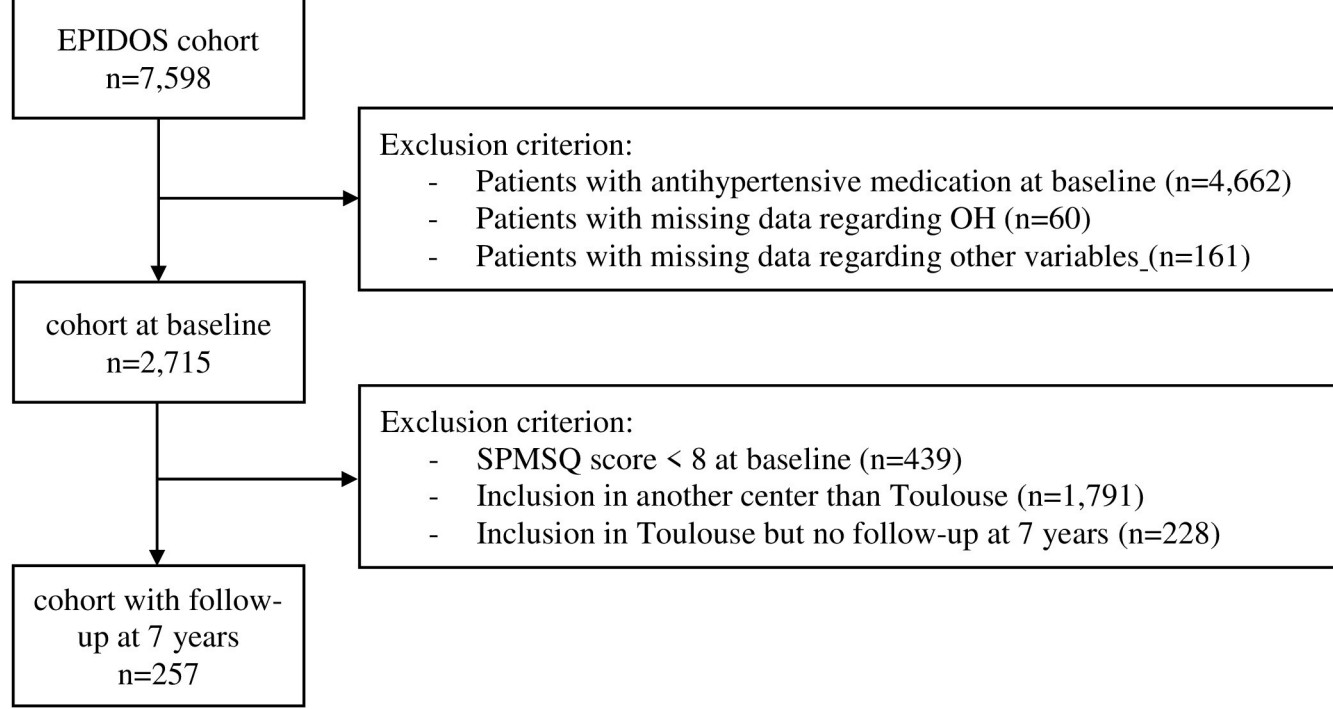

**Fig 1. Flowchart of the study population selection.** EPIDOS, EPIdemiologie de l'OStéoporose; SPMSQ, OH, Orthostatique Hypotension; Short Portable Mental Status Questionnaire.

analysis when they were cognitively impaired at baseline (i.e., Short Portable Mental Status Questionnaire [SPMSQ] score <8; n = 439). Finally, among the 2,715 EPIDOS participants who met the selection criteria at baseline, 257 (9.5%) women with healthy cognitive status at baseline were followed 7 years later in Toulouse. (Fig 1).

### Assessment of orthostatic hypotension

Blood pressure was measured by a nurse in a quiet environment at the inclusion and at the 7-year follow-up. Each measurement was recorded according to a standardized protocol [12] using a sphygmomanometer placed on the brachial artery with the arm at heart level. Blood pressure values were obtained after resting for 15 minutes in the supine position. Global OH was defined as a drop in systolic blood pressure (SBP) $\geq$20 mmHg and/or a drop in diastolic blood pressure (DBP) $\geq$10 mmHg within 3 min after standing [13]. For patients with hypertension, the drop of systolic blood pressure used to define orthostatic hypotension was $\geq$30 mmHg [14, 15]. To seek for possible subtle changes in blood pressures while standing, blood pressure changes were also calculated using the following formulas: $\Delta$SBP in % = (SBP1–SBP2) x 100 / ((SBP1+SBP2)/2) and $\Delta$DBP = (DBP1–DBP2) x100 / ((DBP1+DBP2)/2), where SBP1 and DBP1 were respectively the systolic and diastolic blood pressures measured in supine position, and SBP2 and DBP2 were the systolic and diastolic blood pressure measured after 3 minutes standing.

### Assessment of cognitive performance

Global cognitive function was assessed at baseline using SPMSQ [16], which is a reliable standardized validated screening test for organic brain syndromes. The SPMSQ proved to be a sensitive and specific screening test for moderate to severe dementia for both community-dwellers and hospital inpatients [16]. It consists of a 10-item questionnaire with a score ranging from 0 to 10. The validated cut-off value for normal functioning is a score of 8 or above. In this study, cognitive impairment was consensually defined as a SPMSQ score < 8 [17].

At the seventh year of follow-up, cognitive status was assessed during a single standardized interview using the Mini-Mental State Examination [18] and the Grober and Buschke [19]. The diagnosis of cognitive impairment (i.e., mild cognitive impairment or dementia) was established using the Diagnostic and Statistical Manual of Mental Disorders, fourth edition, criteria [20] in a double-blind manner with the same standardized method of testing by a geriatrician and a neurologist from Toulouse University Hospital, France, who had expertise in dementia. This method has shown excellent inter-rater agreement, with a kappa score close to 1 [21].

### Assessment of covariates

The following covariates were assessed from a physical examination and a health status questionnaire, and included as potential confounders in the statistical models: age, body mass index (BMI), number of comorbidities, pulse pressure (PP), history of stroke, Lawton instrumental activity of daily living (IADL) score, regular physical activity, use psychoactive drugs, and use vitamin D supplements. BMI was calculated as weight/height$^2$ (kg/m$^2$). The number of comorbidities was obtained from a health status questionnaire for the detection of hypertension, diabetes, dyslipidemia, coronary artery disease, chronic obstructive pulmonary disease, peripheral vascular disease, cancer, stroke, Parkinson disease, and depression. Arterial pulse pressure, which is the pressure oscillations around mean pressure, was calculated in millimeters of mercury (mmHg) as the difference between supine SBP and supine DBP. PP is the best marker in older adults of the disproportionate SBP increase reported to DBP [22], which is due in particular to vascular calcification. Functional autonomy was assessed using Lawton

IADL score (/8) [23], with consensual threshold value set at 7. Disability was defined by a score between 0 and 6 out of 8 at the IADL score. Physical activity was considered regular if participants had taken part in at least one recreational physical activity (e.g. walking, gymnastics, cycling, swimming or gardening) for at least 1 hour per week for at least the past month. Finally, medications and vitamin supplementation were reported by direct inquiry. Women were also asked to bring all the medication, including vitamin D supplements, they were regularly taking to the clinical center [24]. Psychoactive drugs were benzodiazepines, antidepressants or neuroleptics.

## Statistical analysis

Characteristics of participants were summarized by means and standard deviations or frequencies and percentages, as appropriate. Firstly, comparisons between participants at baseline categorized into two groups based on cognitive status (cognitive impairment defined as SPMSQ score <8 versus healthy cognitive status defined as SPMSQ score ≥8) were performed using the Chi-square test or the Fisher exact test for qualitative variables, and the Student's $t$-test or the nonparametric Mann-Whitney U test for quantitative variables. Secondly, multiple binary logistic regressions were used to examine the cross-sectional association of cognitive impairment (dependent variable) with ΔSBP and ΔDBP (independent variable) first, and OH then, after adjustment for all studied potential confounders. Thirdly, multiple binary logistic regressions were used to examine the cross-sectional association of cognitive impairment after 7 years of follow-up (dependent variable) with ΔSBP and ΔDBP (independent variable) first, and OH then, after adjustment for all studied potential confounders. P-values<0.05 were considered statistically significant. All statistical analyzes were performed using SPSS (Statistical Package for the Social Sciences; v15.0, IBM Corporation, Chicago, IL).

## Ethics

Women participating in the study were included after having given their written informed consent for research. The study was conducted in accordance with the ethical standards set forth in the Helsinki Declaration (1983). The project was approved by the local ethics committee of each center.

## Results

Among 2,715 community-dwelling older EPIDOS women (mean±standard deviation, 80.1 ±3.7 years), the prevalence of OH was 15.1% and the prevalence of cognitive impairment at baseline was 16.2%. As indicated in Table 1, the OH prevalence did not differ between participants with cognitive impairment at baseline and those without (17.3% versus 14.7% respectively, p = 0.158). There was no significant difference in ΔSBP and ΔDBP variation between the two groups with or without cognitive impairment (1.47±8.22% versus 0.58±8.54% respectively for ΔSBP, p = 0.114; and -2.79±14.70% versus -2.72±11.78% respectively for ΔDBP, p = 0.115). In addition, women with cognitive impairment were older (81.3±4.3 years versus 79.9±3.5 years, p<0.001), had fewer comorbidities (p = 0.007) but higher pulse pressure (p = 0.017), had more often a history of stroke (p = 0.006), had higher prevalence of disability according IADL score (p<0.001), practiced less often a regular physical activity (p = 0.001), and used less often vitamin D supplements (p = 0.033). There was no significant difference for the other characteristics (Table 1).

Multiple logistic regression showed no association between OH and cognitive impairment at baseline (adjusted OR = 1.13, p = 0.388) (Table 2). However, we found a positive association between ΔSBP and cognitive impairment (adjusted OR = 1.01, p = 0.047), but no association

**Table 1. Baseline characteristics according to the cognitive status (n = 2,715).**

| | Total cohort (n = 2,715) | Cognitive status | P-value* | |
| --- | --- | --- | --- | --- |
| | | SPMSQ < 8 (n = 439) | SPMSQ $\geq$ 8 (n = 2,276) | |
| **Clinical measures** | | | | |
| Age (years), mean ± SD | 80.1 ± 3.7 | 81.3 ± 4.3 | 79.9 ± 3.5 | **<0.001** |
| Body mass index (kg/m²) | | | | 0.241 |
| <21: underweight | 463 (17.1) | 86(19.6) | 377(16.6) | |
| [21–25]: Normal | 1051(38.7) | 153(34.9) | 898(39.5) | |
| [25–30]: Overweight | 983(36.2) | 164(37.4) | 819(36.0) | |
| $\geq$30: Obesity | 218(8.0) | 36(8.2) | 182(8.0) | |
| Number of comorbidities [†], mean ± SD | 2.7 ± 1.9 | 2.5 ± 1.8 | 2.7 ± 1.9 | **0.007** |
| Orthostatic hypotension, n (%) | 410 (15.1) | 76 (17.3) | 334 (14.7) | 0.158 |
| ΔSBP (%), mean ± SD | 0.72 ± 8.49 | 1.47 ± 8.22 | 0.58 ± 8.54 | 0.114 |
| ΔDBP (%), mean ± SD | -2.73 ± 12.30 | -2.79 ± 14.70 | -2.72 ± 11.78 | 0.115 |
| Pulse pressure, mean ± SD | 67.9 ± 15.4 | 69.4 ± 15.4 | 67.7 ± 15.4 | **0.017** |
| History of stroke, n (%) | 81 (3.0) | 22 (5.0) | 59 (2.6) | **0.006** |
| Disability, n (%) | 691(25.5) | 191(43.5) | 500(22.0) | **<0.001** |
| Regular physical activity [‡], n (%) | 1455 (53.6) | 203 (46.2) | 1252 (55.0) | **0.001** |
| Use psychoactive drugs [||], n (%) | 1145 (42.2) | 200 (45.6) | 945 (41.5) | 0.117 |
| Use vitamin D supplements, n (%) | 433 (15.9) | 55 (12.5) | 378 (16.6) | **0.033** |

DBP: diastolic blood pressure; IADL: Instrumental Activities of Daily Living score; SBP: systolic blood pressure; SD: standard deviation; SPMSQ: Short Portable Mental Status Questionnaire; *: comparisons based on Chi-square test or the Fisher exact test for qualitative variables, and the Student's t-test or the nonparametric Mann-Whitney U test for quantitative variables, as appropriate; [†]: among hypertension, diabetes, dyslipidemia, coronary heart disease, chronic obstructive pulmonary disease, peripheral vascular disease, cancer, stroke, Parkinson disease and depression; ‡: Instrumental Activities of Daily Living score <7/8; ||: benzodiazepines or antidepressants or neuroleptics; P-value significant (i.e. P<0.05) indicated in bold.

with ΔDBP (adjusted OR = 1.00, p = 0.325). Additionally, cognitive impairment was positively associated with advance in age (p<0.001), and inversely associated with the number of comorbidities (p<0.001) and disability (p<0.001) (Table 2). Using mean arterial pressure value instead of PP as a covariate did not alter our results (data not shown).

Table 3 reports results on the associations between measures of blood pressures at baseline and the incident onset of cognitive impairment after 7 years of follow-up among 257 participants followed in the Toulouse center. We found no association of the cognitive diagnosis after 7 years of follow-up with baseline ΔSBP and ΔDBP (adjusted OR = 0.98 with p = 0.371, and OR = 1.00 with p = 0.345 respectively) and with baseline OH (adjusted OR = 0.53, p = 0.088), adjusted for potential confounders (Table 3).

Of note, the Toulouse group (n = 257) was not fully comparable to the other EPIDOS group of cognitively healthy participants without antihypertensive medication at baseline but not followed (n = 2,019) (S1 Table). In particular, the prevalence of baseline OH (p = 0.002) and the loss of SBP in erect posture (p = 0.034) were higher in the Toulouse group at baseline.

## Discussion

The main finding of the present study is that, although we found no association of OH with cognition, the drop of systolic blood pressure while standing was associated with cognitive impairment at baseline in this large community dwelling older women cohort, irrespective of

**Table 2. Multiple logistic regressions showing the cross-sectional association of baseline cognitive impairment (dependent variable) with ΔSBP and ΔDBP (Model A), and with orthostatic hypotension (Model B) adjusted for potential confounders (n = 2,715).**

| | Baseline cognitive impairment | | | | | |
| --- | --- | --- | --- | --- | --- | --- |
| | Model A | | | Model B | | |
| | OR | [95% CI] | P-value | OR | [95% CI] | P-value |
| Age | 1.06 | 1.03; 1.09 | <**0.001** | 1.06 | 1.03; 1.09 | <**0.001** |
| Body mass index (ref: <21 kg/m$^2$) | | | 0.453 | | | 0.426 |
| [21–25]: Normal | 0.82 | 0.61; 1.10 | | 0.81 | 0.60; 1.10 | |
| [25–30]: Overweight | 0.92 | 0.68; 1.24 | | 0.90 | 0.67; 1.22 | |
| ≥30: Obesity | 0.75 | 0.48; 1.18 | | 0.74 | 0.47; 1.16 | |
| Number of comorbidities [*] | 0.90 | 0.84; 0.95 | <**0.001** | 0.90 | 0.84; 0.95 | <**0.001** |
| Orthostatic hypotension | - | - | - | 1.13 | 0.85; 1.50 | 0.388 |
| ΔSBP | 1.01 | 1.00; 1.03 | **0.047** | - | - | - |
| ΔDBP | 1.00 | 0.99; 1.00 | 0.325 | - | - | - |
| Pulse pressure | 1.00 | 1.00; 1.01 | 0.479 | 1.00 | 1.00; 1.01 | 0.280 |
| History of stroke | 1.51 | 0.90; 2.55 | 0.121 | 1.48 | 0.88; 2.50 | 0.141 |
| Disability [†] | 2.37 | 1.88; 2.98 | <**0.001** | 2.38 | 1.89; 3.01 | <**0.001** |
| Regular physical activity | 0.82 | 0.66; 1.01 | 0.068 | 0.82 | 0.66; 1.02 | 0.081 |
| Use psychoactive drugs [‡] | 1.09 | 0.87; 1.35 | 0.458 | 1.09 | 0.88; 1.35 | 0.442 |
| Use vitamin D supplements | 0.76 | 0.55; 1.04 | 0.081 | 0.76 | 0.55; 1.04 | 0.083 |

DBP: diastolic blood pressure; IADL: Instrumental Activities of Daily Living score; SBP: systolic blood pressure; *: among hypertension, diabetes, dyslipidemia, coronary heart disease, chronic obstructive pulmonary disease, peripheral vascular disease, cancer, stroke, Parkinson disease and depression; †: Instrumental Activities of Daily Living score <7/8; ‡: benzodiazepines or antidepressants or neuroleptics; P-value significant (i.e. P<0.05) indicated in bold. P-value significant (i.e. P<0.05) indicated in bold.

potential confounders. Baseline OH could not predict incident onset of cognitive impairment after 7 years of follow-up.

The association between OH and cognition is the object of growing attention, with conflicting results reported in previous literature [25]. The recent meta-analysis of Isik et al. showed that the risk of finding OH in patients with Alzheimer's disease is multiplied by 2.5 compared to healthy people [26]. Most previous studies reported a positive association, orthostatic hypotension being associated with cognitive impairment or cognitive decline [27–32]. However, a significant number of studies could not report any association of OH with cognitive performance, cerebral blood flow or cerebral damage [33, 34]. Our present results were consistent with these latest findings, but also showed that the mechanism is probably more complex than expected since not the OH *per se*, but the changes in systolic blood pressure were associated with cognition. Thus, our results improve the understanding of the relationship between OH and cognitive impairment in a large cohort of older adults, and are in line with previous studies highlighting an association between systolic orthostatic hypotension and reduced cognitive function [35, 36] or more severe subjective memory complaint in community-dwelling population aged 50 years and over [37]. Finally, as in our study, some other studies found a cross-sectional association of blood pressure drop with cognitive impairment, but not with incident cognitive decline [35, 38–40], just as if the single instantaneous measure of blood pressure change failed to capture a possible long-term effect on neurocognitive health and cognitive decline. Of note, other longitudinal studies showed an association between OH, blood pressure variability or symptoms of OH and incident cognitive decline or incident dementia. However, most of these studies did not exclude participants with antihypertensive medication but only

**Table 3. Multiple logistic regressions showing the cross-sectional association of the cognitive diagnosis at 7 years of follow-up (dependent variable) with ΔSBP and ΔDBP (Model A), and with orthostatic hypotension (Model B) adjusted for potential confounders (n = 257).**

| | Incident onset of cognitive impairment after 7 years of follow-up | | | | | |
|---|---|---|---|---|---|---|
| | Model A | | | Model B | | |
| | OR | [95% CI] | P-value | OR | [95% CI] | P-value |
| Age | 1.21 | 1.10; 1.33 | **<0.001** | 1.17 | 1.08; 1.28 | **<0.001** |
| Body mass index (ref: <21 kg/m$^2$) | | | 0.291 | | | 0.252 |
| [21–25]: Normal | 1.00 | 0.43; 2.34 | | 1.93 | 0.44; 1.97 | |
| [25–30]: Overweight | 1.36 | 0.57; 3.27 | | 1.09 | 0.51; 2.35 | |
| ≥30: Obesity | 2.50 | 0.79; 7.85 | | 2.38 | 0.84; 6.72 | |
| Number of comorbidities * | 0.94 | 0.75; 1.17 | 0.565 | 0.93 | 0.76; 1.13 | 0.464 |
| Orthostatic hypotension | - | - | - | 0.53 | 0.26; 1.10 | 0.088 |
| ΔSBP | 0.98 | 0.94; 1.02 | 0.371 | - | - | - |
| ΔDBP | 0.99 | 0.96; 1.01 | 0.345 | - | - | - |
| Pulse pressure | 0.99 | 0.96; 1.01 | 0.204 | 0.99 | 0.97; 1.01 | 0.410 |
| History of stroke | - | - | - | - | - | - |
| Disability † | 2.47 | 0.93; 6.54 | 0.069 | 2.34 | 1.04; 5.25 | **0.039** |
| Regular physical activity | 0.79 | 0.44; 1.44 | 0.445 | 0.82 | 0.48; 1.40 | 0.457 |
| Use psychoactive drugs ‡ | 1.01 | 0.55; 1.84 | 0.977 | 1.00 | 0.58; 1.73 | 0.988 |
| Use vitamin D supplements | 0.74 | 0.34; 1.62 | 0.456 | 0.74 | 0.35; 1.54 | 0.414 |

DBP: diastolic blood pressure; SBP: systolic blood pressure; *: among hypertension, diabetes, dyslipidemia, coronary heart disease, chronic obstructive pulmonary disease, peripheral vascular disease, cancer, stroke, Parkinson disease and depression; †: Instrumental Activities of Daily Living score <7/8; ‡: benzodiazepines or antidepressants or neuroleptics; P-value significant (i.e. P<0.05) indicated in bold. P-value significant (i.e. P<0.05) indicated in bold.

took the use of antihypertensive medication into account as a covariate [41–43]. Using antihypertensive medication appears however as a major confounding factor and may account for divergences between these previous studies and the present one in which we chose to exclude participants taking these medications. Nevertheless, the differences in prevalence of OH and ΔSBP between the studied sample of participants followed at 7 years and the other participants did not allow extrapolation of these results to the whole EPIDOS cohort and warrant further studies.

How OH, and specifically the decrease in SBP while standing, is related to cognition remains not fully elucidated. First, it is possible that low blood flow pressure resulting from OH may alter brain health and cognitive function. This assumption is in line with the so-called "cardiogenic dementia"; recurrent episodes of cardiac arrhythmias or heart disease exerting a deleterious effect on cognitive performance and mainly on executive functions [44]. Accordingly, previous studies found that OH was associated specifically with executive dysfunction [27, 45]. Second, it is also possible that the neurological disorders manifested by cognitive disorders also result in abnormalities of the autonomic cardiovascular regulation system. Idiaquez et al. reported that the anatomical structures involved in autonomic dysfunction were particularly affected in dementia [46]. Third, another mechanism can be proposed involving the action of neurohumoral mediators such as the norepinephrine, which shows a link with both cognition [47] and OH [48, 49]. Fourth and finally, according to some studies, malnutrition could be involved in the relationship between OH and cognition. A better nutritional status seems to improve global cognition and functional abilities in patients with OH [50].

We also found a significant association between cognitive impairment and advancing age, comorbidity burden and disability in the studied sample of older women. These results are consistent with previous literature [51–53], and confirm the representativeness of our sample and the validity of our finding on the relationship between blood pressures change while standing and cognitive impairment.

The strengths of the present study include the large number of participants, the opportunity to calculate blood pressure changes between supine and standing positions, the possibility of distinguishing systolic and diastolic blood pressures, and the long follow-up of 7 years. Regardless, our study has some limitations. The study cohort was restricted to relatively vigorous older women without antihypertensive medication at baseline who may be unrepresentative of older adults in general. For instance, the prevalence of OH estimated at 15% was lower than in other studies [8]. Thus, the failure to find an association between OH and cognitive impairment may be explained by a lack of statistical power due to the low prevalence of OH in the selected population, or due to the insufficient number of participants despite a nevertheless large sample, with the risk of missing significant differences. Moreover, the study participants may have been more motivated with a greater interest in personal health issues than the general population of older adults. The use of an observational design precludes inferring any causal inference. Orthostatic hypertension was not taken into account as a covariate as such in this study which may partly explain the weak relationship between OH and cognitive, however, the use of delta of variations of SBP and DBP take into account the variations of BP downwards but also upwards which can reflect decubitus hypertension. We used only a single measurement of the BP drop, which may result in nondifferential measurement error and an underestimation of the strength of the true associations. Moreover, it would have been interesting to consider systolic ejection volume and heart rate to evaluate the cardiac output and to obtain information on the autonomous system's activation capabilities during orthostatism. Finally, we took into account as a covariate the use of psychoactive treatments without knowing the details of these treatments used by the patients (i.e., antidepressants, antipsychotics, anxiolytics, etc.) and without having information on the taking of other strongly OH providers including antiparkinsonian drugs. Although, we were able to control for important characteristics that could modify the association between OH and cognition, residual potential confounders such as educational level might still be present. Finally, in the longitudinal analysis, it would have been contributory to assess the persistence of OH after 7 years of follow-up in parallel with the incidental onset of cognitive impairment.

In conclusion, we found that SBP changes while standing, but neither OH *per se* nor DBP drop while standing, were associated with baseline cognitive impairment in older women. In contrast, there was no association of baseline OH with incidental onset of cognitive impairment after 7 years of follow-up. These findings suggest that the relationship between OH and cognitive impairment is more complex than previously expected, and that the drop of SBP specifically may impact adversely cognitive function. Further longitudinal studies with different senior cohorts and taking into account the dysautonomic profile and the severity and persistence of OH during follow-up are warranted to assess a direct effect of systolic OH on cognitive decline.

## Supporting information

**S1 Table. Baseline characteristics of 2,276 participants according to the follow-up at 7 years.**
(DOCX)

## Acknowledgments

Investigators of the EPIDOS study. Coordinators Breart, Dargent-Molina, Meunier, Schott, Hans, and Delmas. Principal investigators: Baudoin and Sebert (Amiens); Chapuy and Schott (Lyon); Favier and Marcelli (Montpellier); Hausherr, Menkes and Cormier (Paris); Grandjean and Ribot (Toulouse).

## Author Contributions

**Conceptualization:** Guillaume T. Duval, Cedric Annweiler.

**Data curation:** Anne-Marie Schott, Yves Rolland.

**Formal analysis:** Guillaume T. Duval, Jennifer Gautier, Cedric Annweiler.

**Funding acquisition:** Anne-Marie Schott, Yves Rolland.

**Investigation:** Cedric Annweiler.

**Methodology:** Guillaume T. Duval, Cedric Annweiler.

**Project administration:** Guillaume T. Duval, Cedric Annweiler.

**Resources:** Yves Rolland.

**Software:** Jennifer Gautier, Cedric Annweiler.

**Supervision:** Cedric Annweiler.

**Validation:** Anne-Marie Schott, Yves Rolland, Hubert Blain, Gustavo Duque, Cedric Annweiler.

**Writing – original draft:** Guillaume T. Duval.

**Writing – review & editing:** Guillaume T. Duval, Cedric Annweiler.

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
