## [Decision Letter · Decision Letter 0]

13 Jul 2022

PONE-D-22-15462Orthostatic hypotension and neurocognitive disorders in older women: Results from the EPIDOS cohort studyPLOS ONE

Dear Dr. Duval,

Thank you for submitting your manuscript to PLOS ONE. After careful consideration, we feel that it has merit but does not fully meet PLOS ONE’s publication criteria as it currently stands. Therefore, we invite you to submit a revised version of the manuscript that addresses the points raised during the review process.

The manuscript does not reach enough level for acceptance in the journal. See the reviewers’ suggestions carefully and respond to them appropriately.

We look forward to receiving your revised manuscript.

Kind regards,

Masaki Mogi

Academic Editor

PLOS ONE

Journal Requirements:

2. PLOS requires an ORCID iD for the corresponding author in Editorial Manager on papers submitted after December 6th, 2016. Please ensure that you have an ORCID iD and that it is validated in Editorial Manager. To do this, go to ‘Update my Information’ (in the upper left-hand corner of the main menu), and click on the Fetch/Validate link next to the ORCID field. This will take you to the ORCID site and allow you to create a new iD or authenticate a pre-existing iD in Editorial Manager. Please see the following video for instructions on linking an ORCID iD to your Editorial Manager account: https://www.youtube.com/watch?v=_xcclfuvtxQ.

Reviewers' comments:

Reviewer's Responses to Questions

**Comments to the Author**

1. Is the manuscript technically sound, and do the data support the conclusions?

Reviewer #1: Partly

Reviewer #2: Yes

2. Has the statistical analysis been performed appropriately and rigorously? 

Reviewer #1: Yes

Reviewer #2: Yes

3. Have the authors made all data underlying the findings in their manuscript fully available?

Reviewer #1: No

Reviewer #2: Yes

4. Is the manuscript presented in an intelligible fashion and written in standard English?

Reviewer #1: Yes

Reviewer #2: Yes

5. Review Comments to the Author

Reviewer #1: I reviewed the manuscript entitled “Orthostatic hypotension and neurocognitive disorders in older women: Results from the EPIDOS cohort study.” It is very interesting with a large sample size but has many concerns that need to be considered.

1. All the abbreviations should be defined.

2. Considering that almost half of the elderly patients use antihypertensive drugs, and the patients taking antihypertensive drugs were also not included in this study, it seems that the results are far from reflecting real life and cannot be generalized.

3. Furthermore, given the fact that hypertensive patients were included in the study, in such patients should have been evaluated with a cut-off of 30 mm Hg fall in SBP, which may be more appropriate, since blood pressure fall after standing up largely depends on baseline BP ( https://pubmed.ncbi.nlm.nih.gov/29562291/ , https://pubmed.ncbi.nlm.nih.gov/30900378/ )

4. Literature review in the MN needs to update since many recent studies, in which OH was demonstrated that it may be related to dementia, have been overlooked. (https://pubmed.ncbi.nlm.nih.gov/34255194/
https://pubmed.ncbi.nlm.nih.gov/33975062/
https://pubmed.ncbi.nlm.nih.gov/33735870/ . Please discuss them.

5. All medications that can cause OH in your study population, such as antiparkinson medication, antidepressants, and antipsychotics, should have been evaluated in the study.

All these concerns have to be clarified and mentioned as limitation

Reviewer #2: In the manuscript, Duval et al. have reported the relationship between “Orthostatic hypotension and neurocognitive disorders in older women in the EPIDOS cohort study". This study is interesting. However, there were several critical flaws.

Major comments

#1: The power analysis

Did the authors perform the power analysis? If not, there was some possibility that the results were derived by chance.

#2: n=257

It was complex to undestand how the authors reached the n=257. It would be helpful if there was the flow chart for the way of reaching to n=257.

#3: Pulse pressure

Pulse pressure might be adjusted in the regression model. How the systolic, diastolic or mean BP was adjusted instead for pulse pressure?

#4: Orthostatic hypertension

Did the authors take into account for orthostatic hypertension? This might be confounded in the weak relationship between OH and cognitive dysfunction in the prospective study.

#5: Cognitive decline

How would be the result if the dependent variable was defined as delta cognitive function (Baseline score minus score at 7yrs) instead of cognitive dysfunction?

6. PLOS authors have the option to publish the peer review history of their article (what does this mean?). If published, this will include your full peer review and any attached files.

Reviewer #1: No

Reviewer #2: **Yes: **Michiaki Nagai

---

## [Author Response · Author response to Decision Letter 0]

6 Oct 2022

We appreciated the comments offered by the Reviewer. We replied point by point to the Reviewer’s comments in the document "Response to the Reviewers" and performed a revision of the manuscript. All changes have been highlighted on the revised manuscript, and a final manuscript without highlighting was also updated.

---

## [Decision Letter · Decision Letter 1]

25 Oct 2022

PONE-D-22-15462R1Orthostatic hypotension and neurocognitive disorders in older women: Results from the EPIDOS cohort studyPLOS ONE

Dear Dr. Duval,

Thank you for submitting your manuscript to PLOS ONE. After careful consideration, we feel that it has merit but does not fully meet PLOS ONE’s publication criteria as it currently stands. Therefore, we invite you to submit a revised version of the manuscript that addresses the points raised during the review process.

The manuscript needs a response to the Reviewer's comments.See the comments carefully and respond them appropriately. 

We look forward to receiving your revised manuscript.

Kind regards,

Masaki Mogi

Academic Editor

PLOS ONE

Reviewers' comments:

Reviewer's Responses to Questions

**Comments to the Author**

1. If the authors have adequately addressed your comments raised in a previous round of review and you feel that this manuscript is now acceptable for publication, you may indicate that here to bypass the “Comments to the Author” section, enter your conflict of interest statement in the “Confidential to Editor” section, and submit your "Accept" recommendation.

Reviewer #1: All comments have been addressed

Reviewer #2: All comments have been addressed

2. Is the manuscript technically sound, and do the data support the conclusions?

Reviewer #1: Yes

Reviewer #2: Yes

3. Has the statistical analysis been performed appropriately and rigorously? 

Reviewer #1: Yes

Reviewer #2: Yes

4. Have the authors made all data underlying the findings in their manuscript fully available?

Reviewer #1: Yes

Reviewer #2: Yes

5. Is the manuscript presented in an intelligible fashion and written in standard English?

Reviewer #1: Yes

Reviewer #2: Yes

6. Review Comments to the Author

Reviewer #1: I have reviewed the revised MN entitled "Orthostatic hypotension and neurocognitive disorders in older women: Results from the EPIDOS cohort study".

Many thanks to the authors, as all the revisions made are sufficient.

Reviewer #2: The manuscript was substantially revised.

Major comments

#1: Adjusting for absolute SBP or DBP

Pulse pressure was adjusted in the regression model. However, the systolic, diastolic or mean BP should be at least adjusted instead for pulse pressure. These results should be provided.

7. PLOS authors have the option to publish the peer review history of their article (what does this mean?). If published, this will include your full peer review and any attached files.

Reviewer #1: No

Reviewer #2: **Yes: **Michiaki Nagai

---

## [Author Response · Author response to Decision Letter 1]

24 Jan 2023

We thank Reviewers for the second review of our manuscript. As suggested, we have performed a revision of our manuscript. All the changes are highlighted in yellow on the revised manuscript. 

Please see ou Response to Reviewers document to find our point-by-point responses to the comments offered by the Reviewer. 

Major comments

Reviewer #1: Adjusting for absolute SBP or DBP

Pulse pressure was adjusted in the regression model. However, the systolic, diastolic or mean BP should be at least adjusted instead for pulse pressure. These results should be provided.

We thank Reviewer #1 for pointing out this important point.

We have taken this comment into account in the analysis of the results. By replacing, as proposed, Pulse Pressure (PP) by Mean Blood Pressure (MBP) while lying down at rest, in the analysis models, we do not observe any modification of the results.

These data from Table 2 are presented below but have not been added to the manuscript for clarity. These results have simply been described in the Results section. “Using mean arterial pressure value instead of PP as a covariate did not alter our results (data not shown)”.

Please see correction page 11 line 223-224.

---

## [Decision Letter · Decision Letter 2]

27 Jan 2023

Orthostatic hypotension and neurocognitive disorders in older women: Results from the EPIDOS cohort study

PONE-D-22-15462R2

Dear Dr. Duval,

We’re pleased to inform you that your manuscript has been judged scientifically suitable for publication and will be formally accepted for publication once it meets all outstanding technical requirements.

Kind regards,

Masaki Mogi

Academic Editor

PLOS ONE

Additional Editor Comments (optional):

Reviewers' comments:

Reviewer's Responses to Questions

**Comments to the Author**

1. If the authors have adequately addressed your comments raised in a previous round of review and you feel that this manuscript is now acceptable for publication, you may indicate that here to bypass the “Comments to the Author” section, enter your conflict of interest statement in the “Confidential to Editor” section, and submit your "Accept" recommendation.

Reviewer #2: All comments have been addressed

2. Is the manuscript technically sound, and do the data support the conclusions?

Reviewer #2: Yes

3. Has the statistical analysis been performed appropriately and rigorously? 

Reviewer #2: Yes

4. Have the authors made all data underlying the findings in their manuscript fully available?

Reviewer #2: Yes

5. Is the manuscript presented in an intelligible fashion and written in standard English?

Reviewer #2: Yes

6. Review Comments to the Author

Reviewer #2: The manuscript was well revised. And the authors addressed well. No further questions were there from myside.

7. PLOS authors have the option to publish the peer review history of their article (what does this mean?). If published, this will include your full peer review and any attached files.

Reviewer #2: **Yes: **Michiaki nagai

---

## [Editor Report · Acceptance letter]

15 Feb 2023

PONE-D-22-15462R2 

Orthostatic hypotension and neurocognitive disorders in older women:
Results from the EPIDOS cohort study 

Dear Dr. Duval:

I'm pleased to inform you that your manuscript has been deemed suitable for publication in PLOS ONE. Congratulations! Your manuscript is now with our production department. 

Kind regards, 

on behalf of

Dr. Masaki Mogi 

Academic Editor

PLOS ONE